# Anisotropic resistance with a 90° twist in a ferromagnetic Weyl semimetal, Co$_2$MnGa

Nicholas P. Quirk [1], Guangming Cheng [2], Kaustuv Manna [3], Claudia Felser [3], Nan Yao [2] & N. P. Ong [1] ✉

Weyl semimetals exhibit exotic magnetotransport phenomena such as the chiral anomaly and surface-to-bulk quantum oscillations (Weyl orbits) due to chiral bulk states and topologically protected surface states. Here we report a unique transport property in crystals of the ferromagnetic nodal-line Weyl semimetal Co$_2$MnGa that have been polished to micron thicknesses using a focused ion beam. These thin crystals exhibit a large planar resistance anisotropy (10×) with axes that rotate by 90 degrees between opposite faces of the crystal. We use symmetry arguments and electrostatic simulations to show that the observed anisotropy resembles that of an isotropic conductor with surface states that are impeded from hybridization with bulk states. The origin of these states awaits further experiments that can correlate the surface bands with the observed 90° twist.

In a Weyl semimetal (WSM), the breaking of time-reversal invariance (TRI) or inversion symmetry leads to the splitting of each Dirac node into two Weyl nodes that are separated in **k** space and have opposite chiralities. The surface projections of these Weyl nodes act as terminations for topologically protected Fermi arcs[1]. Transport experiments on single crystals have observed the chiral anomaly involving bulk states in parallel electric and magnetic fields[2–5] and Fermi-arc-mediated transport has been inferred through quantum oscillations in thin crystals[6–10]. However, so far novel transport phenomena in WSMs have only been observed when topological states are brought into the quantum limit at low temperatures in a strong magnetic field.

In a WSM that is ferromagnetic, the breaking of TRI leads to interesting features. Inhomogeneities in the magnetization **M** such as magnetic domain walls are predicted to host localized charge and equilibrium currents that may produce unusual transport behavior[11–13]. Motivated by effects arising from the breaking of TRI, we studied the transport properties of thin (0.4–5 µm) plate-like crystals of Co$_2$MnGa, a nodal-line WSM and room-temperature ferromagnet[14]. These platelets host just a few magnetic domains. The domain walls form a regular pattern (Fig. 1d) but are easily erased by a weak applied field. We uncover a set of anomalous transport properties in these thin crystals that, unexpectedly, are

unaffected by strong magnetic fields and therefore appear to lie beyond domain-wall effects.

All six crystals investigated display a large conductance anisotropy on both of the broad faces (the $a$–$b$ plane). However, the principal axes of the anisotropy are invariably rotated by 90° between the upper and lower surfaces (Fig. 1). Using symmetry arguments and simulations, we infer that the anisotropies originate from surface states $|\mathbf{q}\rangle$ that are protected from hybridization with bulk states $|\mathbf{k}\rangle$, i.e., surface-to-bulk charge transfer is mediated by a transfer matrix $t(\mathbf{q}, \mathbf{k})$ that is strongly anisotropic. The intrinsic 90° twist impedes charge flow in a way reminiscent of how crossed polarizers block light transmission (but the physics is different). The obstruction leads to transport anomalies that satisfy an explicit roto-inversion symmetry $C_4I$, and are observable up to macroscopic length scales (5 µm) at room temperature.

## Results

Co$_2$MnGa forms a face-centered cubic lattice, space group $Fm\bar{3}m$ (no. 225) and is a soft ferromagnet ($H_C \simeq 30$ Oe) with Curie temperature $T_C = 690$ K. Two majority-spin bands near the Fermi energy form Weyl nodal lines that are twofold degenerate due to mirror symmetry. The nodal lines, as well as drumhead surface states, have been observed by photoemission[14]. Additionally, Co$_2$MnGa exhibits giant anomalous Hall

[1]Department of Physics, Princeton University, Princeton, NJ 08544, USA. [2]Princeton Institute for the Science and Technology of Materials, Princeton University, Princeton, NJ 08544, USA. [3]Max Planck Institute for Chemical Physics of Solids, Nöthnitzer Str. 40, 01187 Dresden, Germany. ✉e-mail: npo@princeton.edu

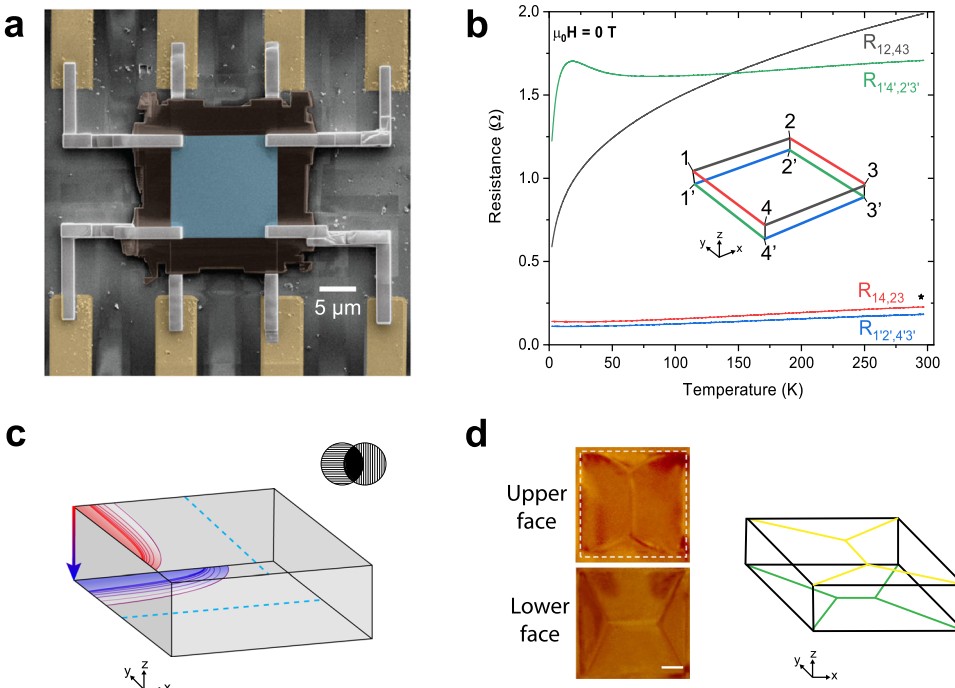

**Fig. 1 | Co₂MnGa thin lamella devices. a** Top-view false-color SEM micrograph of Sample N2 ($12 \times 12 \times 0.4$ µm). The central blue region is the thin Co₂MnGa crystal. Insulating, amorphous carbon (dark brown) is added by FIB to protect the sides of the device. Electrical contacts at all 8 corners are made by depositing platinum strips (gray) via FIB and connecting them to pre-patterned gold electrodes (yellow). **b** The temperature dependences of the four-point resistances on the upper/lower faces ($H = 0$ T) in Sample O1 (thickness $= 1$ µm). The high-conductance-axis (upper $= \hat{\mathbf{y}}$, lower $= \hat{\mathbf{x}}$) resistances have metallic temperature dependences that match those of bulk Co₂MnGa crystals. (The isotropic-equivalent resistance is denoted with a black star at $T = 290$ K.) The low-conductance-axes (upper $= \hat{\mathbf{x}}$, lower $= \hat{\mathbf{y}}$) resistances have anomalous temperature dependences that decay slowly at high temperatures but sharply below $T = 10$ K. The inset diagram depicts the corner-contact labeling scheme used throughout to identify each $R_{ij,kl}$. **c** A diagram depicting the 90°-twist planar anisotropy in a 3D lamella. The dashed blue lines indicate the high-conductance axes. The anisotropy is such that when current is directed along the $z$-axis edges, a surface potential arises that is strongly distorted in opposite lateral directions on the upper/lower faces. This effect is reminiscent of how crossed polarizers block light (upper-right inset). **d** The left inserts are magnetic force micrographs of the remanent magnetic domains observed in thin Co₂MnGa lamellae ($5 \times 5 \times 1.3$ µm) (without electrical contacts). These feature Landau flux-closure patterns rotated by 90° between upper and lower faces. Scale bar $= 1$ µm. The sketch on the right depicts the arrangement of the surface magnetization textures on the 3D lamellae.

and Nernst responses that have been related to the large Berry curvature at the Weyl nodal lines[14–18].

The present transport experiments were motivated by images of anomalous remanent magnetic domain patterns obtained by magnetic force (MFM) microscopy. Using a focused ion beam (FIB) microscope, we cut thin square-shaped lamellae of thickness $d \lesssim 1$ µm from as-grown Co₂MnGa single crystals (see Methods). Each sample contains just a few domains that form the familiar Landau flux-closure pattern[19] on each face (Fig. 1d). However, the pattern on the lower face is rotated by 90° with respect to that of the top. This 90° twist remains in lamellae with $d$ down to 100 nm. (An imaging study of the twisted remanent domain-wall patterns will be reported by G. C. and N. Y.)

Initially, we tried to identify the anisotropies in the resistance matrix $R_{ij,kl}$ (defined below) with conductances along domain walls. (We do not know if the twisted flux-closure pattern locks to a high- or low-conductance axis; MFM cannot be done on lamellae with electrical contacts.) However, Co₂MnGa is a soft ferromagnet with a coercive field of just 30 Oe. We find that magnetotransport measurements on lamellae equipped with electrical contacts show clear evidence for the erasure of domain walls at low field. As shown below, the Hall resistances feature sharp hysteretic peaks at the coercive field and saturate to a constant slope at 1.3 T. Furthermore, each $R_{ij,kl}$ has very weak magnetoresistance: <5% of the zero-field value at $R_{ij} = 3$ T. Because all elements in $R_{ij,kl}$ are nearly field independent, we eventually concluded that the observed anisotropies are intrinsic to the twisted electronic band structure rather than arising from domain walls.

We investigate the electrical anisotropies by measuring the non-local resistance matrix $R_{ij,kl} = V_{kl}/I_{ij}$ where $V_{kl}$ is the voltage measured between contacts ($k, l$) with current applied to contacts $i$ and $j$. We label the four vertices of the upper (lower) face as $1 \cdots 4$ ($1' \cdots 4'$), as shown in the inset of Fig. 1b. All resistance measurements are true four-probe ($i, j$ are distinct from $k, l$). To distinguish the anomalous, intrinsic anisotropies from trivial geometrical effects, we compare the measured $R_{ij,kl}$ against the values calculated for a metallic slab (of identical shape and size) with the isotropic resistivity of bulk Co₂MnGa crystals (130 µΩ·cm at 290 K[14]). The calculation was performed using the software package COMSOL. We refer to this as the isotropic equivalent. Our present analysis is focused on measurements of one device (Sample O1) with dimensions $10 \times 10 \times 1$ µm and edges that are defined with precise alignment to the facets of the source Co₂MnGa crystal. We align our coordinate system with the observed planar anisotropy: the $\hat{\mathbf{x}}$ ($\hat{\mathbf{y}}$) axis is defined by the high-conductance axis in the lower (upper) face in each sample. For additional sample details see Sections A, B, and C in the Supplementary Information (SI).

Figure 2a depicts the resistance anisotropy that arises when current is directed in the $x–y$ plane. All measurements are at $T = 290$ K and $H = 0$ T. With the pair of directed current contacts $ij$ (written as the vector $\overrightarrow{ij}$) $\parallel \hat{\mathbf{y}}$ on the upper face, the resulting voltage in that plane agrees with the expected value for the isotropic equivalent: $R_{14,23} = 0.2$ Ω (experimental) ≈ 0.29 Ω (isotropic equivalent). However, with $\overrightarrow{ij} \parallel \hat{\mathbf{x}}$, the resistance is ~10× greater: $R_{12,43} = 1.9$ Ω. Yet, in both configurations the voltages on the lower face remain largely undistorted from their expected isotropic values (blue arrows in Fig. 2a). If we direct the

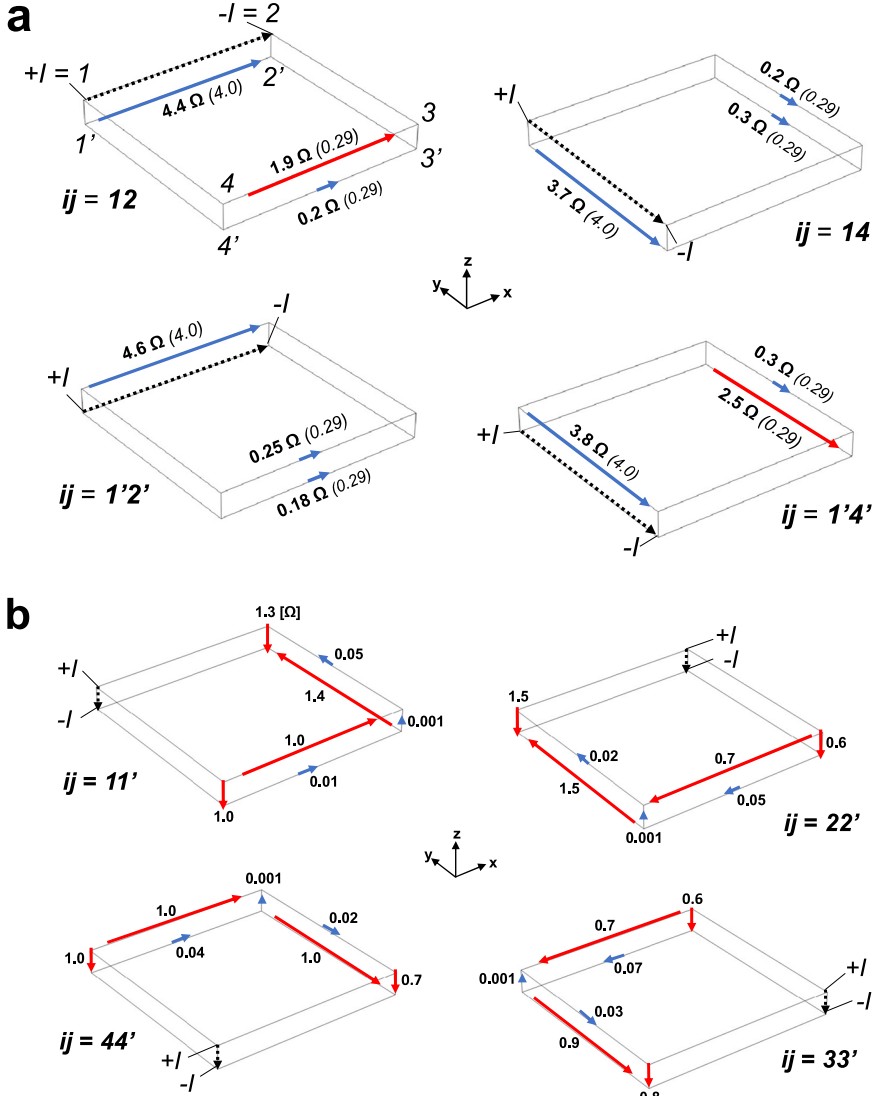

**Fig. 2 | 90°-twist resistance anisotropy. a** Four-point resistances measured in Sample O1 at 290 K ($H = 0$ T) with the current directed in the $x$–$y$ ($a$–$b$) plane. Each panel is identified by the source and drain current contacts ($ij$) according to the corner-labeling scheme shown in the upper-left panel. For each $I_{ij}$, the voltage drop along every other parallel edge, $V_{kl}$, is measured and the resulting $R_{ij,kl} = V_{kl}/I_{ij}$ is recorded in Ohms. The length and color of the arrows represent the magnitude of the resistance and the agreement with the isotropic equivalent, respectively. The blue arrows match the isotropic-equivalent values whereas the red arrows indicate $R_{ij,kl}$ that are anomalously large. The isotropic-equivalent values are given in parenthesis. When the current is directed along a high-conductance axis (upper = $\hat{\mathbf{y}}$, lower = $\hat{\mathbf{x}}$), all of the resulting $R_{ij,kl}$ agree with the isotropic equivalent (blue arrows). **b** When the current is directed along the 1-μm-wide $z$-axis edges, a highly distorted electrical potential landscape arises. The expected values in the isotropic equivalent are all $\simeq 10^{-7} \Omega$. Thus, the red arrows correspond to resistances that are anomalously large by 6 orders of magnitude. The blue arrows indicate near-zero resistance. Vanishing of the potential difference across the vertical edge (e.g. 33′) diametrically opposite to the current contacts (11') is consistent with the $C_4 I$ symmetry inherent to the 90° twist (Eq. (2)).

current in the lower face, a planar anisotropy arises that is rotated by 90°. With $\overrightarrow{ij} \parallel \hat{\mathbf{x}}$, $R_{1'2',4'3'} = 0.18\ \Omega$ agrees with the isotropic equivalent, but $R_{1'4',3'2'} = 2.5\ \Omega$ ($\parallel \hat{\mathbf{y}}$) is an order of magnitude larger. Thus, the crystal has a unique resistance anisotropy with a high conductance axis $\parallel \hat{\mathbf{y}}$ ($\hat{\mathbf{x}}$) on the upper (lower) face. When current is directed along the high-conductance axes, all of the $R_{ij,kl}$ agree with the values predicted by the isotropic equivalent. Only when the current is directed along a low-conductance axis is an anomalously large resistance observed, and then only along the edge parallel to the current axis on that face.

Out-of-plane currents ($\overrightarrow{ij} \parallel \hat{\mathbf{z}}$) bring out the most dramatic features of the anisotropy (Fig. 2b). In the isotropic equivalent, the 7 distinct $R_{ij,kl}$ for $I$ applied along each vertical edge are $\simeq 10^{-7} \Omega$ (consistent with the non-local resistance in a high-conductivity metal). However, in the experiment four of these resistances are 6 orders of magnitude greater, at about 1 Ω. For example, with $ij = 11'$ (top-left

panel in Fig. 2b) the voltages on the adjacent vertical edges are 1.3 Ω ($kl = 22'$) and 1.0 Ω ($kl = 44'$). Along the diametrically opposed edge ($kl = 33'$), however, the resistance is very small, about −1 mΩ. Additionally, the in-plane voltages alternate between ≈1 Ω and ≈ 10 mΩ−on edges of the crystal that are separated by just 1 μm.

Together these measurements map out a surface potential $V(\mathbf{r}_s)$ with a robust 90°-twist symmetry. The diagrams in each of the four panels of Fig. 2b (corresponding to $\overrightarrow{ij}$ along each vertical edge) can be mapped into each other by a rotation about the $z$ axis by an angle $\phi = n\pi/2$ ($n \in \mathbb{Z}$) followed by an inversion about the origin (geometric center of the crystal). Thus, we identify the symmetry group of the observed 90° twist as $C_{4z} I$. This procedure can also be applied to the in-plane resistance configurations shown in Fig. 2a.

The 90°-twist anisotropy is largest at room temperature but still present at 2 K. Additionally, there is a marked difference in the

temperature dependences of the low- and high-conductance-axis resistances (Fig. 1b). The high-conductance axis resistances, $R_{14,23}$ and $R_{1'2',4'3'}$, have metallic profiles matching bulk Co$_2$MnGa. The low-conductance-axis resistances, however, have anomalous temperature dependences. $R_{12,43}$ decays gradually at high temperature but sharply below 10 K and $R_{1'4',2'3'}$ has a non-monotonic dependence with a peak at 10 K. Similar temperature dependences have been observed in other samples (see Section B in the SI). The temperature dependences reveal that there is an additional weakly broken symmetry between the two low-conductance-axis resistances. However, the (sub-dominant) mechanism that breaks this symmetry is unknown.

As mentioned, the anisotropies are virtually unaffected by the magnetic field (see Section D in the SI). With $\mathbf{H} \parallel \hat{\mathbf{z}}$, we measured the magnetoresistance (MR) and Hall effect using the contacts on both faces of the crystal. Figure 3a shows the field dependences of each of the four in-plane $R_{ij,kl}$ depicted in Fig. 1b ($T$ = 200 K). Each curve is extremely flat. In Fig. 3b we subtract the zero-field resistance from each curve in order to resolve the very weak MR ($\leq$ 10 m$\Omega$) that occurs at low field (Sample N2, thickness = 400 nm). Supplementary Fig. 3a shows a similar plot for Sample O1. At $|H|$ ~ 0.75 T, there are local maxima in the MR $\parallel \hat{\mathbf{y}}$ and minima in the MR $\parallel \hat{\mathbf{x}}$. Similar ($x$-$y$) anisotropic MR have been observed in Co$_2$MnGa thin films where they have been attributed to domain-dependent scattering[20].

An additional distinguishing feature in the MR is that the $R_{ij,kl}$ with $\vec{ij} \parallel \hat{\mathbf{y}}$ are dominantly antisymmetric in $H$ (Fig. 3b, c). In fact, taking the field-antisymmetric component of these curves results in field dependences that look remarkably like the anomalous Hall response,

with the same knee at $|H|$ = 1.3 T (Supplementary Fig. 3b). Furthermore, Fig. 3 shows how changing the current and voltage contacts changes the sign of the antisymmetric slope, $R_{ij,kl}(\mathbf{H}) = R_{kl,ij}(-\mathbf{H})$, in excellent agreement with the reciprocity theorem (Fig. 3c)[21]. The origin of this antisymmetric MR, which appears similarly in each sample, is unknown to us. In Section D of the SI we describe additional instances of anti-symmetric MR that arise when $\mathbf{H}$ is directed in the $x$–$y$ plane (Supplementary Fig. 4).

The Hall-effect curves reveal an unusual effect arising from the 90°-twist anisotropy. On each face, we measure the field profiles of the four transverse resistances: $R_{13,42}$ and $R_{42,31}$ on the upper face, and $R_{1'3',4'2'}$ and $R_{4'2',3'1'}$ on the lower. (In our sign convention $R_{ij,kl} > 0$ for holes in an isotropic square plate, implying $V_{kl} > 0$ with current source at $i$ and drain at $j$.) We find that, on the upper face, $R_{13,42}$ is comprised of a large $H$-independent (background) term $\Delta R$ that dwarfs its conventional $H$-antisymmetric profile. Surprisingly, its rotated partner $R_{42,31}$ shows the same weak $H$-antisymmetric term but its $H$-independent term is now negative (cyan curves in Fig. 3d for Sample O1 at 290 K). On the lower face, $R_{1'3',4'2'}$ and $R_{4'2',3'1'}$ exhibit $H$-antisymmetric terms closely similar to those in the upper face but they are now strongly shifted by large background terms $\Delta R'$ multiplied by an additional negative sign (gray curves). We may understand the origin of $\Delta R$ and $\Delta R'$ (including their signs) by examining the equipotential curves that arise in simulations where the planar conductivity anisotropy is converted into the geometrical stretching of an isotropically conducting slab. On the upper face (left insert diagram in Fig. 3d), the resistance anisotropy can be modeled with an isotropically conducting rectangle

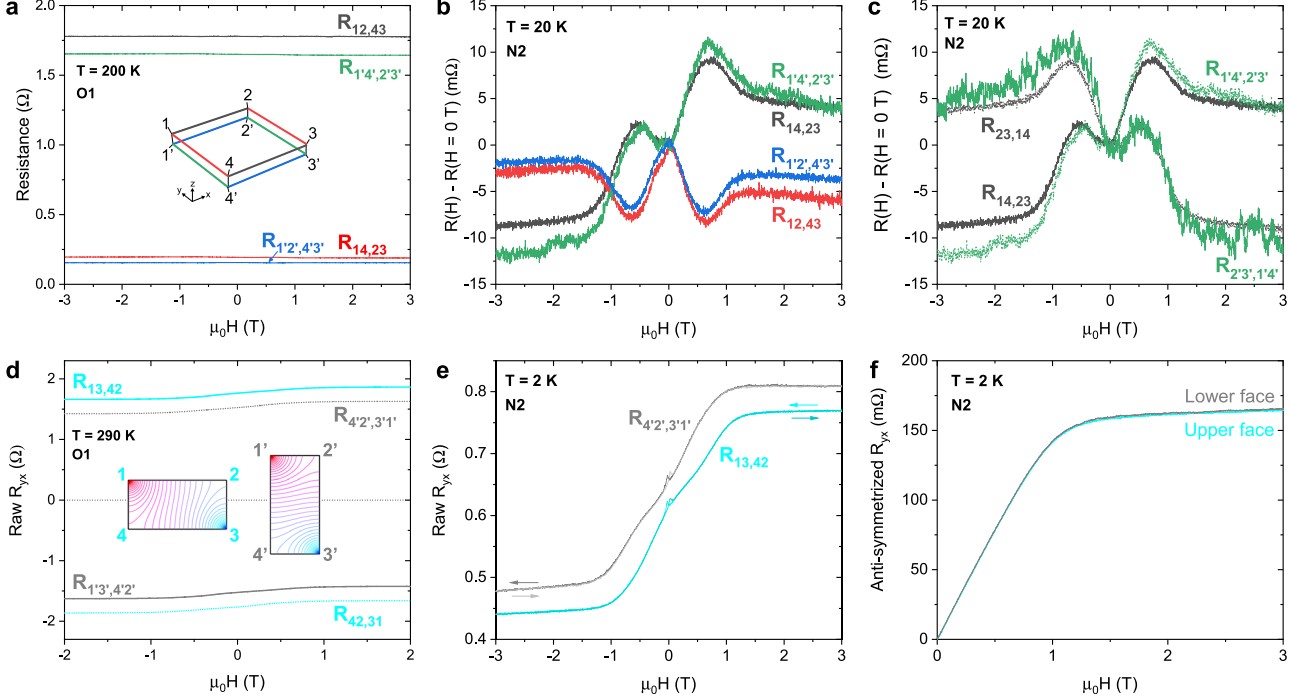

**Fig. 3 | Magnetoresistance and Hall effect (H ∥ ẑ). a** The field dependences of the four resistances shown in Fig. 1b at $T$ = 200 K. **b** The magnetoresistance (MR) of each in-plane resistance for Sample N2 ($T$ = 20 K). The zero-field values have been removed in order to resolve the weak MR of each channel. These are: $R_{14,23}$ = 0.31 Ω, $R_{12,43}$ = 1.71 Ω, $R_{1'4',2'3'}$ = 1.89 Ω, and $R_{1'2',4'3'}$ = 0.48 Ω. Each MR is featureless above $H$ = 1.3 T. Below 1.3 T, the MR ∥ŷ (∥x̂) exhibit local maxima (minima) at $|H|$ = 0.75 T. Additionally, the MR ∥ŷ are dominated by an $H$-antisymmetric component. **c** Exchanging the current and voltage contacts to opposite edges of the sample, $R_{ij,kl} \leftrightarrow R_{kl,ij}$, inverts the sign of these antisymmetric MR. **d** This panel depicts the transverse (Hall) configurations, $R_{ij,kl}$ with $\vec{ij} \perp \vec{kl}$, measured on the upper (cyan) and lower (gray) faces. The curves have the same $H$ dependence but background offsets of differing signs. The opposite planar anisotropies in each face of the

crystal can be mapped onto isotropic rectangles that are stretched along opposite $x$–$y$ axes. The insert diagrams show simulated equipotential contours on these equivalent isotropic rectangles for $R_{13,42}$ (left) and $R_{1'3',4'2'}$ (right). Due to the opposite stretching, with $I_{13}$, $V_{42} > 0$ and with $I_{1'3'}$, $V_{4'2'} < 0$. This is the origin of the positive (negative) backgrounds in $R_{13,42}$ ($R_{1'3',4'2'}$). $R_{42,31}$ and $R_{4'2',3'1'}$ develop opposite signs in their background terms by the same mechanism. **e** High-resolution field sweeps of the transverse resistances (Sample N2, $T$ = 2 K) show sharp hysteretic peaks at the coercive field (30 Oe), indicative of the weak pinning of magnetic domains. **f** The intrinsic Hall response, derived by adding together the transverse $R_{ij,kl}$ measured in each face. The Hall response measured using the set of contacts on the upper face (cyan) is identical to that of the lower face (gray).

that is lengthened along the less conducting axis $\hat{\mathbf{x}}$. The sign of the potential drop $V_{42} > 0$ translates to a positive $H$-independent $\Delta R$ in $R_{13,42}$. On the lower face, the isotropic equivalent rectangle is longer along $\hat{\mathbf{y}}$ (right insert). The sign of $V_{3'1'}$ (also positive) translates to a positive $\Delta R'$ in $R_{4'2',3'1'}$. For the rotated partners on either face, the signs invert. Hence we see that the large $H$-independent terms are inherent to the 90° twist. They cancel when we average the two curves on either face to yield the intrinsic Hall response $R_{yx}$ (Fig. 3f).

Figure 3e shows high-resolution scans of the two transverse resistances with positive offsets, $R_{13,42}$ and $R_{4'2',3'1'}$, measured in Sample N2 at 2 K. Small, hysteretic peaks can be resolved at the coercive field (30 Oe). These provide additional evidence of the very weak pinning of magnetic domains in these samples. Figure 3f shows that after averaging the two transverse $R_{ij,kl}$ and anti-symmetrizing by $H$, the Hall response measured on each face is identical. Furthermore, after scaling by the lamella thickness (400 nm), the anomalous Hall resistivity $\rho_{yx}(H = 3\,\text{T}) = 160\,\text{m}\Omega$, agrees with that of the as-grown (isotropic) source crystals (6 μΩ-cm at 2 K)[14].

The 90°-twist anisotropy arises in every thin crystal we have studied, including samples as thick (thin) as 5 μm (0.4 μm), with a 2:1 aspect ratio, and with edges oriented at a 45° angle with respect to the crystal lattice. (See Supplementary Table S4 in the SI for a full summary.) Although there is some variation in the magnitude of the anisotropy between samples, each has high-conductance-axis resistances that agree with the expected values for the equivalent isotropic slabs and low-conductance-axis resistances that are fixed at the few-Ohm level.

## Discussion

### Symmetry constraints

Once we discovered empirically that $R_{ij,kl}$ satisfies $C_4I$ symmetry, the number of configurations needed to characterize the anisotropy was reduced considerably. With current applied out-of-plane ($\vec{ij} \parallel \hat{\mathbf{z}}$) there are 7 distinct configurations, whereas for planar current $\vec{ij} \parallel \hat{\mathbf{x}}, \hat{\mathbf{y}}$ there are 14 (within the symmetry group). In the planar configurations, we further exclude the four transverse voltage pairs which may be inferred by applying the closed-loop theorem. Hence 13 measurements suffice to characterize $R_{ij,kl}$.

Symmetry leads to a more interesting constraint on out-of-plane transport. Arguably, the most singular feature is the appearance of near-zero resistance on the vertical edge diametrically opposite to the current contacts, which is juxtaposed between extremely large resistances across the two remaining vertical edges. For e.g. with current $I_{11'}$, $V_{33'} \simeq 0$ whereas both $R_{11',22'}$, $R_{11',44'} \simeq 1\,\Omega$. The vanishing $V_{33'} \simeq 0$ is a consequence of the $C_4I$ symmetry independent of a microscopic model.

We consider the electric potential function $V(\mathbf{r}_s)$ on the two side surfaces 233'2' and 344'3' with the current injected at 1 and drained at 1' ($\mathbf{r}_s$ locates a point on either side surface). For convenience, we map $\mathbf{r}$ onto the 2D plane with coordinates $(u,z)$ and origin at the mid point of 33'. The $C_4I$ symmetry implies that, on the $(u,z)$ plane, the potential distribution must satisfy the two constraints (more details are given in Section E of the SI)

$$V(u,z) = -V(-u,-z), \quad V(u,z) \neq V(-u,z). \tag{1}$$

The first is a consequence of $C_{2d}$ (rotation by $\pi$ of the $(u,z)$ plane about the normal through its origin). The second is a consequence of current flow in a potential field. With the assumed current, $V$ on the upper edge 32 must be higher than that on the lower edge 3'2' (Section E in the SI). The simplest function satisfying Eq. (1) has the form

$$V(u,z) = u(z \cdot \text{sign}(u) - d/2). \tag{2}$$

Hence symmetry constrains $V$ to vanish all along the edge 33' ($u = 0$).

### 3D Simulation

A discussion of the approaches we adopted to understand the measured $\{R_{ij,kl}\}$ is given in Section F of the SI. We derived the most insights from finite-element electrostatic simulations using COMSOL. In loose analogy with crossed polarizers in optics, we may regard the 90° twist as a mechanism that obstructs charge transport along $\hat{\mathbf{z}}$ with observable effects over macroscopic length scales (sample thickness $d$ up to 5 μm). This seems physically possible only if the surface states $|\mathbf{q}\rangle$ in Co$_2$MnGa are protected against hybridization with the bulk states $|\mathbf{k}\rangle$ and charge transfer between the surface and the bulk proceeds by a hopping matrix $t(\mathbf{q}, \mathbf{k})$. We express the anisotropic conductivity as

$$\begin{aligned} \sigma_{xx}^u &= \sigma_0/\alpha, \quad \sigma_{yy}^u = \sigma_0 \\ \sigma_{xx}^l &= \sigma_0, \quad \sigma_{yy}^l = \sigma_0/\alpha \quad (\alpha > 1), \end{aligned} \tag{3}$$

for the upper and lower surfaces, respectively. The transfer matrix $t(\mathbf{q}, \mathbf{k})$ should reflect this anisotropy as well. (For example, if the anisotropy is expressed as an anisotropic surface band mass, $m_y \sim m_0$ and $m_x = \alpha m_0$ where $m_0$ is the bulk mass, Fermi wavevector mismatch strongly suppresses the transfer rate for $\mathbf{q} \parallel \hat{\mathbf{x}}$.)

Heuristically, we may simulate the effect of an anisotropic $t(\mathbf{q}, \mathbf{k})$ by replacing the strictly 2D surface states by an ultrathin layer of a 3D anisotropic conductor that has a conductivity tensor $\sigma_{ij}^u = \text{diag}[\sigma_0/\alpha, \sigma_0, \sigma_0/\beta]$ (upper surface) with $\beta \gg \alpha > 1$. When $\beta$ is very large ($10^4 - 10^5$), the vanishing $z$-axis conductivity simulates an overall weak amplitude for the transfer matrix.

Using the Electric Currents COMSOL package, we represent the crystal as a lamellar slab of size $10 \times 10 \times (1-2\delta)$ μm and isotropic conductivity $\sigma_0$, sandwiched between two ultrathin sheets of conductors with conductivity $\sigma_{ij}^{u(l)}$ and thickness $\delta$. For $\sigma_0$ we used the value $7.7 \times 10^5\,(\Omega\,\text{m})^{-1}$, measured in bulk crystals at 290 K[14]. We enforce continuity of the electric field across the interfaces and add point-contact probes to each of the 8 vertices. With specific choices of $I_{ij}$ and $V_{kl}$, we then simulate $R_{ij,kl}$ by calculating the voltage drops $V(\mathbf{r})$ across the edges of the composite slab. We choose $\alpha$ and fine-tune $\beta$ to match the experimental low-conductance-axis resistance $R_{12,43} = 1.9\,\Omega$ within 0.5%. We then evaluate how well the remaining $R_{ij,kl}$ outputs agree with the experimental values. Due to the $C_4I$ symmetry it suffices to simulate only $\frac{1}{4}$ of the possible four-point resistance configurations. Although convergence problems preclude simulations with $\delta < 10$ nm, we find that this picture can recreate the observations with surprising accuracy (the best agreement is achieved in the limit $\delta = 10$ nm).

This simple model captures the key aspects of the observed 90°-twist anisotropy in these thin crystals. With $\delta = 10$ nm, we run the simulation with $\alpha = \{10, 100, 500, 1000\}$ and record the resulting $R_{ij,kl}$ (Fig. 4a, b). The $\beta$ required to match the experimental low-conductance-axis resistance ($R_{14,23}$) for each $\alpha$ are $\{5.76, 4.31, 3.65, 3.4\} \times 10^4$, respectively. With these ($\alpha$, $\beta$), the model succeeds in holding the high-conductance-axis resistances to their experimental values, which match the isotropic equivalent, while also generating anomalous resistances that agree with the observed 90° twist. It nearly captures the large 1-Ω voltage drops for current directed along the thin $z$-axis edges, reaching $\simeq 0.8\,\Omega$ for $\alpha = 1000$. Furthermore, it preserves the vanishingly small resistances that arise in this configuration, which are enforced by the $C_4I$ symmetry. It fails to capture the full magnitude of the voltage drop directly beneath the applied-current edge, undershooting $R_{14,1'4'}$ and $R_{12,1'2'}$ by ~50%, however. As $\alpha$ is increased, the agreement improves, but still falls short ($\alpha = 1000$ is close to the asymptotic value). This issue could perhaps be rectified by a more elaborate model in which the role of the narrow side-wall faces of the crystal is taken into account.

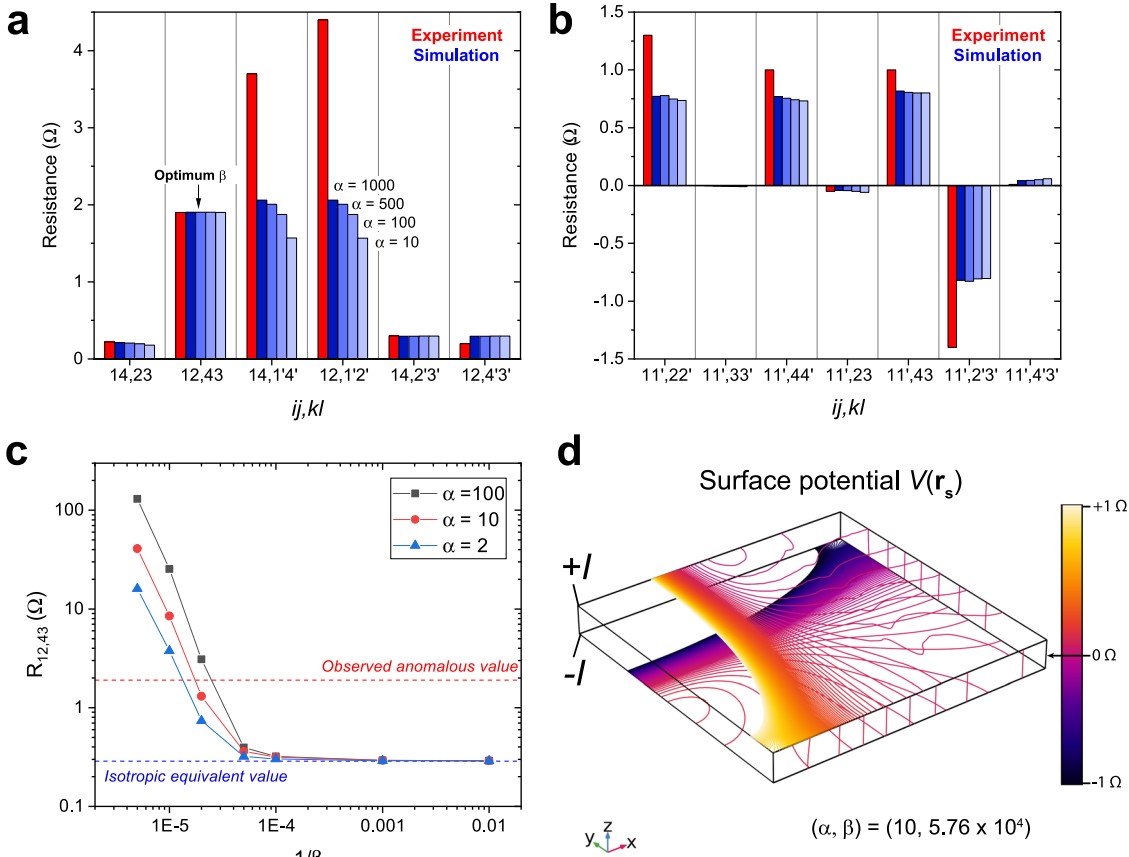

**Fig. 4 | Simulation results for a model with ultrathin anisotropic surfaces.**
**a**, **b** Comparison of the experimental (red) and simulated (shades of blue) resistances in the $\delta = 10$ nm thin-surface model. For a range of $\alpha = \{10, 100, 500, 1000\}$, $\beta$ was tuned so that the simulated $R_{12,43}$ (low-conductance axis) matched the experimental resistance within 0.5%. The optimal $\beta$ for these $\alpha$ were found to be 5.76, 4.31, 3.65, and $3.4 \times 10^4$, respectively. Panel a (b) presents the results with the current contacts $\vec{ij} \parallel \hat{\mathbf{x}}, \hat{\mathbf{y}} (\parallel \hat{\mathbf{z}})$. Due to the $C_4I$ symmetry, it suffices to simulate only $\frac{1}{4}$ of the possible four-point resistance configurations. **c** This plot shows that although the experimentally observed anomalous resistances can be recreated for a range of

$\alpha$, no anisotropy arises for $\beta < 10^4$. Without a large suppression of the out-of-plane conductivity in the surfaces, the anomalous resistances revert to the isotropic-equivalent values. **d** This diagram shows the surface potential contours that arise when current is directed $\parallel \hat{\mathbf{z}}$, $ij = 11'$. The range has been limited to $\pm 1$ $\Omega$ about the $C_4I$-symmetric point at the midpoint of the $33'$ edge in order to resolve the contours that extend across the sample. The spacing between contours is 0.01 $\Omega$. They appear to have discrete jumps between the two faces because the thickness of the ultrathin anisotropic surfaces cannot be resolved (1% of the total slab thickness).

The simulation results are only weakly dependent on $\alpha$, but critically sensitive to the out-of-plane anisotropy $\beta$. For example, the simulated $R_{ij,kl}$ that arise for $\alpha = 10$ are within 10% of those for $\alpha = 1000$. However, we find that no anisotropy arises for $\beta < 10^4$ (Fig. 4c). Below this critical value, the surface low-conductance-axis resistances are short-circuited by the isotropic bulk. Therefore, the surface conductances must be partially isolated from the bulk in order to generate the observed 3D twisted anisotropy. This is most evident when the current is applied out-of-plane ($\vec{ij} \parallel \hat{\mathbf{z}}$). The 1-$\Omega$ resistances on neighboring $z$-axis edges are a consequence of the strong planar distortion of the current in the ultrathin surfaces. Figure 4d depicts the highly distorted $V(\mathbf{r})$ contours that arise in a simulation that agrees with the experimental results. Due to the large $\beta$, with the source contact at 1, the current density cannot easily flow directly to the drain at 1′. A portion spreads out laterally in the surfaces before finding the path of least resistance through the bulk. The large magnitudes of the voltage drops at 22′ and 44′ are thus a measure of the weak amplitude of the transfer matrix $t(\mathbf{q}, \mathbf{k})$ for surface states to enter the bulk.

As the thickness of the anisotropic surface sections is increased, the simulated $R_{ij,kl}$ largely diverge from the experimental values. Figure 5a, b show the results of increasing the surface thickness $\delta$ with $\alpha$ fixed at 100. Again, the optimal $\beta$ values are determined by enforcing agreement with the resistance $R_{12,43} = 1.9$ $\Omega$ along the low-conductance axis. As $\delta$ is increased (10, 100, 300, 500 nm), the picture

transitions from one with two anisotropically conducting surface planes to one with a large portion of the current passing through the bulk. (The $\delta = 500$ nm model consists of just two anisotropic slabs). In these thicker-surface models two key conditions are not met: the high-conductance-axis resistances are increased away from the isotropic-equivalent values, and the $z$-axis anomalous resistances drop from ~1 $\Omega$. Increasing $\delta$ systematically drives the simulation away from the extremely planar experimentally observed anisotropy to a milder one that fails to agree for both the high- and low-conductance axes. In fact, the requirement that the high-conductance-axis resistances, e.g., $R_{14,2'3'}$ and $R_{12,4'3'}$, match the isotropic-equivalent values is quite stringent. In order for them to agree, isotropic states with the conductivity of bulk $Co_2MnGa$ must occupy most of the sample volume.

## Implications and outlook

Initially, our working assumption was that the anisotropies observed in $R_{ij,kl}$ arise from current paths confined to domain walls. However, as discussed above, we abandoned this viewpoint following the Hall and magnetoresistance measurements which show that all $R_{ij,kl}$ values remain nearly unchanged (at the few % level) in $H$ as large as 9 T (unsuccessful models are described in Section F of the SI). Our conclusion is that the anisotropies in $R_{ij,kl}$ are intrinsic to the electronic band structure when subject to the 90° twist. The emergence of this twist in all lamellar crystals investigated shows that it reflects an

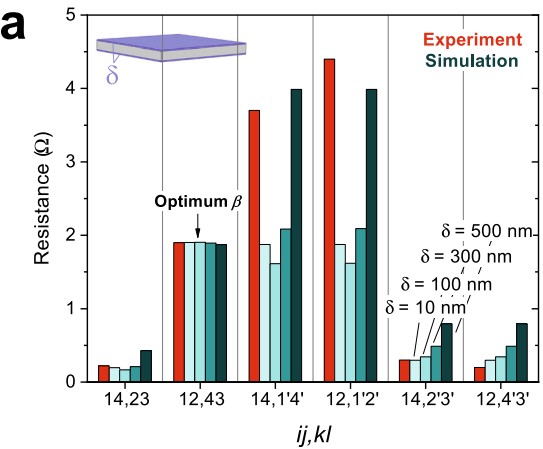
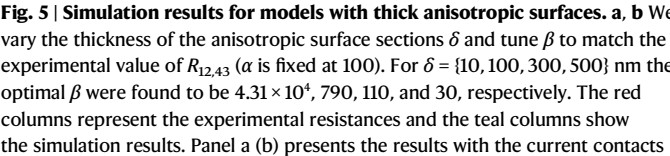
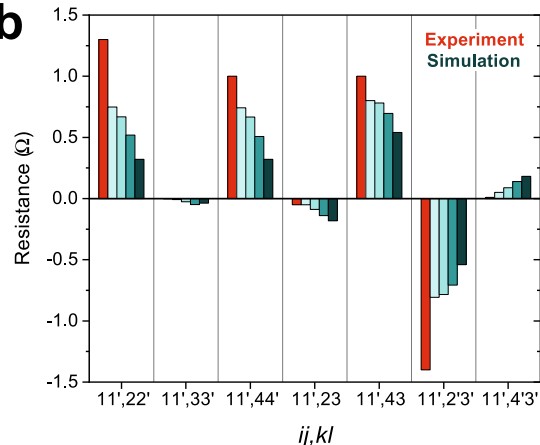

**Fig. 5 | Simulation results for models with thick anisotropic surfaces. a, b** We vary the thickness of the anisotropic surface sections $\delta$ and tune $\beta$ to match the experimental value of $R_{12,43}$ ($\alpha$ is fixed at 100). For $\delta = \{10, 100, 300, 500\}$ nm the optimal $\beta$ were found to be $4.31 \times 10^4$, 790, 110, and 30, respectively. The red columns represent the experimental resistances and the teal columns show the simulation results. Panel a (b) presents the results with the current contacts

$\vec{ij} \parallel \hat{x}, \hat{y}$ ($\parallel \hat{z}$). As $\delta$ is increased, the simulation results largely diverge from the experimentally observed anisotropy. In particular, the high-conductance-axis configurations ($ij = 14$) no longer match those of an isotropic slab−a key feature of the experimental anisotropy−and the z-axis non-local resistance anisotropy becomes much less extreme.

intrinsic instability in Co₂MnGa. In effect, the bulk electronic bands spontaneously undergo the 90° twist in thin crystals.

These unusual effects raise a number of open questions. What is the microscopic mechanism that underlies this rare instability? Is the twist driven by the topologically non-trivial nature of the bulk states? Where is the energy gain that offsets the cost incurred by the twist? What gives rise to the large conductance anisotropy of the protected surface states, and why does the Landau magnetic domain pattern lock to the conductance anisotropy axes?

Quite apart from these questions, we have in these crystals an opportunity to detect the presence of anisotropic surface states that are protected from hybridizing with high-conductance isotropic bulk states. We observed that the 90°-twist impedes the out-of-plane charge current in a complex way consistent with the underlying $C_4I$ symmetry. For current applied to say the vertices $11'$, the voltage drops across both adjacent vertices $22'$ and $44'$ are extremely large ($R_{11',22'} \sim 1$ Ω) whereas that across the opposite vertices, $33'$, is extremely small ($R_{11',33'} \sim 1$ mΩ). Although these anisotropies are reproducible in our simulated models, the underlying physics for the unexpected appearance of 1-Ω non-local resistances in a metallic crystal of sub-micron thickness requires further investigation.

## Methods

### Crystal growth

Single crystals of Co₂MnGa were grown using the Bridgman-Stockbarger growth technique. First, we prepared a polycrystalline ingot using the induction melt technique with the stoichiometric mixture of Co, Mn, and Ga metal pieces of 99.99% purity. Then, we poured the powdered material into an alumina crucible and sealed it in a tantalum tube. The growth temperature was controlled with a thermocouple attached to the bottom of the crucible. For the heating cycle, the entire material was melted above 1200 °C and then slowly cooled below 900 °C. The L2₁ crystal structure and quality were confirmed by X-ray diffraction and transmission electron microscopy (TEM).

### Thin lamella device fabrication

Thin Co₂MnGa lamellae were prepared by focused Ga²⁺ ion beam milling using a ThermoFisher Scientific Helios NanoLab™ 600 dual-beam system (FIB/SEM) and standard TEM sample preparation techniques. First, a thick slab (~10 × 10 × 10 μm) is cut from the source Co₂MnGa crystal and lifted out with an in situ micro-manipulator

needle. The sample is secured to the needle via platinum deposition using the gas-injection system (GIS) in the FIB/SEM chamber. Next, a 3-μm-thick amorphous carbon protection layer is added to each side face using the GIS. The sample is then affixed to a copper lift-out grid with platinum so that it can be manually rotated to on edge-on orientation with the ion beam. It is then gently polished to the desired thickness (~1 μm) using a 2 kV ion excitation voltage and beam currents of <100 pA over a few hours. The beam is held at a fixed (edge-on) direction to the slab and scanned line-by-line from the outside-in, first at the edge of the upper face and then the lower face. The beam direction is the same in both faces. The grazing incidence angle and low ion excitation energy significantly minimize gallium implantation and dislocation defects in the lamella surfaces.

We make electrical connections to all 8 corners of the lamellae by depositing platinum with the GIS. We create the electrodes in two steps. First, we deposit 1-μm-thick platinum "nibs" at the corners on the backside of the lamella. We then flip the sample over (mounted onto the lift-out grid) and carefully use the micro-manipulator needle to maneuver it into place on the SiO₂ substrate. During the subsequent steps we take care not to image the sample directly with the ion beam. We then fix the sample to the substrate with a small amount of platinum at a corner of the carbon barrier and then cut it away from the needle with the ion beam. We then deposit platinum strips to connect the nibs, which extend out to the sides, to nearby gold electrodes pre-patterned on the chip by photolithography. To make the four contacts on the upper face, we first build up stacks of platinum strips in a staircase geometry that abut the carbon barrier. We then connect the staircases across to the crystal corners. The amorphous carbon layer electrically insulates the upper-face platinum electrodes from touching the sides of the crystal. In order to minimize the resistances of these connections, we build thick (1-μm) strips with the ion-beam scanning direction parallel to the long axis and use the shortest lengths possible. With this technique, the contact resistances in our devices are typically about 200 Ω.

### Magnetic force microscopy

Magnetic domains were imaged using a Bruker ICON3 atomic force microscope equipped with Bruker's MFM probes (MESP-V2). The MFM probe was coated with Co-Cr for high sensitivity and magnetic contrast. The probe had a nominal tip radius of 35 nm, a nominal resonant frequency of 75 kHz, a nominal coercivity of 400 Oe (medium) and a

magnetic moment of $1 \times 10^{-13}$ emu (medium). Before the experiment, the MFM probe was magnetized under an external magnetic field (top-down) for 15 min.

## Electrical transport

Resistance measurements were carried out in a Quantum Design (QD) Physical Property Measurement System (PPMS) equipped with a 9 T magnet. A Keithley 6221 current source (100-500 µA) and 2182a nano-voltmeter in Delta mode were used to record each four-point resistance. There was no observed difference between single-direction (DC) and alternating (AC) current pulses. Measurements with in-plane field were accomplished using the QD Horizontal Rotator option in the PPMS.

## Data availability

The figure source data generated in this study have been deposited in the Figshare database under accession code https://doi.org/10.6084/m9.figshare.22806593.

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

## Acknowledgements

The authors have benefited from discussions with L. Balents, B.A. Bernevig, J. Herzog-Arbeitman, and B. Lian. The research was funded by the U.S. Department of Energy (DE-SC0017863) and the Gordon and Betty Moore Foundation's EPiQS initiative via grant GBMF9466 (to N.P.O.). This research made use of the Imaging and Analysis Center operated by the Princeton Materials Institute at Princeton University, which is supported in part by the Princeton Center for Complex Materials, a National Science Foundation Materials Research Science and Engineering Center (Award DMR-2011750). This publication was supported by the Princeton University Library Open Access Fund.

## Author contributions

N.P.Q., G.C., N.Y., and N.P.O. conceptualized and designed the experiment. Device fabrication was carried out by G.C. and N.P.Q. N.P.Q. performed all electrical measurements as well as the finite-element simulations. The crystals were grown by K.M. and C.F. Analyses of the data were carried out by N.P.Q. and N.P.O. who jointly wrote the manuscript with input from all authors.

## Competing interests

The authors declare no competing interests.
