## [Peer Review File · Nature Communications]

Reviewers' Comments:

Reviewer #1:

Remarks to the Author:

The manuscript reported systematically transport measurements on thin Co₂MnGa lamellar, cut from single-crystals using a focused ion beam. These lamellar crystals exhibit unexpected and highly unusual planar resistance anisotropy with principal axes that rotate by 90 degree between the upper and lower faces. Using symmetry arguments and simulations, they infer that the anisotropies originate from surface states that are protected from hybridization with bulk states, i.e., surface-to-bulk charge transfer is mediated by a transfer matrix that is strongly anisotropic. This idea sounds interesting and the experimental data seems to support their suggestion. However, all of these transport results actually cannot provide very solid evidence, as mentioned by authors that the origin of these states awaits further experiments that can correlate the surface bands with the observed 90°-twist geometry. In the last part of the manuscript, the authors also raised many open questions on the microscopic mechanism that underlies this rare instability, they ruled out the viewpoint of the anisotropies observed arises from current paths confined to domain walls. I have a number of concerns about their experiments.

1) In Figure 1b, the resistivity of the crystal exhibits very clear anisotropy, but why in Figure 3a and 3b, the MR shows no any anisotropy (upper face: $R_{12,43}$, lower face: $R_{1'2',4'3'}$)?

2) Why the MR, $R_{14,23}$, $R_{1'4',2'3'}$, in Figure 3 are non-symmetric, what is the physics?

3) Since the thickness of the crystal is very thin, only about a few micrometer, how to avoid the top electrodes not touching the bottom surface or vice versa? I understand this is a very tough experiment, does all of the six samples show exactly the same features?

Reviewer #2:

Remarks to the Author:

The manuscript by N. Quirk et al. reports the observation of an unusual type of anisotropy in the electrical transport of a ferromagnetic Weyl semimetal, where the 90-degree rotation between the high-conductance axes of the top and bottom surfaces is documented, reproduced in multiple samples, and found to be consistent with C_{4i} symmetry. The weak magnetic field dependence of this anisotropy up to 9T, in contrast to the elimination of ferromagnetic domain walls at around 30Oe, suggests that this unusual behavior is independent of the domain wall conductance, but intrinsic to the electronic structure of the material. It is postulated to arise from surface states protected from strong hybridization with bulk states, and to support this hypothesis, an electrostatic finite element simulation has been used to attempt to reproduce the experimental results in a model consisting of a slab sandwiched between two ultrathin layers of anisotropic conductivity. To the best of my knowledge, no similar experimental results have been reported before, and it is a rather novel type of observation that serves a broad interest in the field of correlated electrons and topological materials. The experimental and analytical methods used are well established and explained. I would like to recommend this paper for publication in Nature Communication, but before doing so, I have some concerns that I hope the authors will address.

1. It is not clear to me how the x and y axes are defined and how this definition can be consistent across different samples and for comparison between electrical transport and MFM (Fig. 1d). The sample has cubic symmetry, while the x and y axes are along crystallographically equivalent directions (Table S4). In addition, according to the manuscript, the 90-degree rotation should be a spontaneous one reflecting the electronic instability (line 359). This suggests to me that it is equally possible for the top surface to have a high-conductance axis along x and along y, as well as for the bottom surface.

(1) However, in all samples measured, it is consistently reported that the top surface (with vertices 1,2,3,4) has high conductance along y, while the bottom surface (with vertices 1',2',3',4') has high conductance along x. Do the x and y axes have physical differences, or is it just a matter of practice to maintain consistent notation?

(2) If it is a matter of notation, why do the R vs. T curves (Figure 1b and Figure S1) always show $R_{12,43}$ as monotonic while $R_{1'4',2'3'}$ is non-monotonic? This observation seems to suggest that surfaces with x- and y-directions as their high-conductance axes, or top and bottom (in contact with the SiO₂ substrate) ARE different, but why should they be?

(3) For the MFM image shown in Figure 1d, if there is no in situ electrical transport measurement, how is it determined whether the Landau flux closure pattern is aligned with the high or low conductance axes? Alternatively, if there is always an EXPLICIT symmetry breaking between x- and y-directions or top and bottom surfaces, should any mechanisms be identified?

2. The contact issue described in the Supplementary Materials, or whatever is causing the absence of low conductance in certain configurations, can be a major concern that potentially undermines the quality of the experiment. The absence of Ohm-scale resistance in some configurations of sample N2 has been attributed to poor contact 3' (lines 77-85 in the Supplementary Texts). However, in Tables S2 and S3, configurations such as {1'4', 2'3'}, {2'3', 1'4'}, {33', 22'}, {22', 33'}, {11', 2'3'} all involve contact 3' as either a current or voltage contact, and the greater-than-IsoEquiv resistance has all been identified. Why do the authors think that contact 3' specifically fails the measurements at {1'4', 33'}, {2'3', 44'}, {33', 44'}, {33', 1'4'}, {44', 33'}, and {44', 2'3'}? Could there be other experimental problems? Will these factors affect the reported results in other configurations?

3. In lines 185-189 it is mentioned that "although the raw transverse resistances, e.g. $R_{\{13,42\}}$ and $R_{\{4'2',3'1'\}}$, reflect the intrinsic 90 deg rotation anisotropy at zero H", I understand that the authors are trying to attribute the raw resistance difference between $R_{\{13,42\}}$ and $R_{\{4'2',3'1'\}}$ to the 90 deg rotation, but this connection is not obvious. In fact, at zero field, if a rotation of $\pi/2$ is performed followed by an inversion with respect to the crystal center (C_{4I} as identified in the manuscript), the configuration {13,42} is exactly mapped to {4'2',3'1'}. Do the authors mean that it is the fact that they are inequivalent that reflects the anisotropy? Then I don't follow the argument. Nevertheless, these planar transverse channels are not discussed at all in other parts of the manuscript, such as Figures 1, 2, 4, 5 and Tables S1, S2 and S3. They deserve a clearer explanation if the authors want to establish the trivial field-dependence of the anisotropy.

4. The simulation results presented in Figure 4 and Figure 5 do not seem to me to be dramatically different in their comparability to the experimental results, especially considering how the experimental results can vary in different samples (Table S4). Although I still think the COMSOL simulation is a valuable analysis, its support for the presence of surface states may not be as strong as one hopes. Even if the authors still want to advocate the surface state hypothesis, which I do not disagree with, I think alternative perspectives should also be offered.

5. There are a few minor problems with the manuscript.

(1) Figure 1b: It is not noted in which sample these T-dep curves are measured.

(2) Figure 2 has been referred to as Figure 3 throughout the manuscript and in the supplementary materials.

(3) Lines 253-254 give only the conductivity matrix for the top surface, not that of the bottom surface. I assume this is just a 90-degree rotation, but it should be noted.

(4) Figure 4d: Shouldn't there be a color bar indicating how the color scheme manifests the gradient of $V(r)$? I noticed that the caption says that the spacing between the contours is 0.01Ohm, but this does not help in the regions where the density of contours is high.

(5) For the COMSOL simulation results shown in Figure 5, neither the caption nor the main text states what the optimal beta values are. This makes it difficult to compare with the results in Figure 4 and to evaluate whether the model used for the simulation remains reasonable when the delta increases dramatically.

(6) Figure S1: It is said that "the abrupt jump at $T = 120K$ in $R_{\{1'4',2'3'\}}$ was not seen in repeated cool-downs", then I suggest that a normal curve from one of the "repeated cool-downs" be shown instead of this one, which shows identified artifacts.

(7) Table S4: Sample F1 also has an in-plane aspect ratio of 2:1, I am puzzled why the isotropic equivalent values for x and y directions do not differ, as they do in sample J1. Sample F1 is discussed in Supplementary Texts D2 for orientation dependence, where this aspect ratio difference is not mentioned.

(8) Figure S4: In panel e, I suspect that the green curve should be $R_{\{12,1'2'\}}$ ($H//x$) instead of $R_{\{12,2'2'\}}$ ($H//x$) as labeled. I also wonder why its counterpart, i.e. $R_{\{12,1'2'\}}$ with $H//y$, is not shown in the same way as all the other three configurations.

(9) Figure S5: 4'-3'-3'-2 should be 4'-3'-3-2.

Response to reviewers comments

REVIEWER COMMENTS

Reviewer #1 (Remarks to the Author):

The manuscript reported systematically transport measurements on thin Co₂MnGa lamellar, cut from single-crystals using a focused ion beam. These lamellar crystals exhibit unexpected and highly unusual planar resistance anisotropy with principal axes that rotate by 90 degrees between the upper and lower faces. Using symmetry arguments and simulations, they infer that the anisotropies originate from surface states that are protected from hybridization with bulk states, i.e., surface-to-bulk charge transfer is mediated by a transfer matrix that is strongly anisotropic. This idea sounds interesting and the experimental data seems to support their suggestion. However, all of these transport results actually cannot provide very solid evidence, as mentioned by authors that the origin of these states awaits further experiments that can correlate the surface bands with the observed 90°-twist geometry. In the last part of the manuscript, the authors also raised many open questions on the microscopic mechanism that underlies this rare instability, they ruled out the viewpoint of the anisotropies observed arises from current paths confined to domain walls. I have a number of concerns about their experiments.

Response to Reviewer #1:

Thank you very much for taking the time to review our paper, we have addressed each of your questions to the best of our ability below.

1)In Figure 1b, the resistivity of the crystal exhibits very clear anisotropy, but why in Figure 3a and 3b, the MR shows no anisotropy (upper face: $R_{12,43}$, lower face: $R_{1'2',4'3'}$)?

In the (old version) Figs. 3a and 3b the zero-field resistances had been removed in order to highlight the very small MR (< 10 mΩ) that arises for $H < 1.3$ T (mentioned in the old caption). In the new version we have made the subtraction of the zero-field resistances more explicit; instead of labeling the y axes as “magnetoresistance,” they are labeled “ $R(H) - R(H = 0 \text{ T})$ ” in the new figures. Furthermore, we have added a new panel (Fig. 3a) that highlights how minuscule the MR is compared to the intrinsic twisted anisotropy (in each curve the total change in MR at 9 T is less than 20 mΩ, whereas the intrinsic anisotropy is $\sim 2 \Omega$).

2)Why the MR, $R_{14,23}$, $R_{1'4',2'3'}$, in Figure 3 are non-symmetric, what is the physics?

We do not know why $R_{14,23}$ and $R_{1'4',2'3'}$ (MR || \mathbf{y}) are predominantly antisymmetric in H . We devoted a section (E) in the Supplementary Information to this, showing how the H-antisymmetrized components of the MR closely resemble the anomalous Hall effect (Fig. S3b).

3)Since the thickness of the crystal is very thin, only about a few micrometers, how to avoid the top electrodes not touching the bottom surface or vice versa?

Using the focused-ion beam (FIB) gas injection system (GIS) we can fabricate micron-sized structures on the sample out of both carbon (insulating, 100 k Ω – > M Ω depending on thickness) and platinum (conducting). See section A.2 in the Supplementary Materials (SM) for a full description of our thin lamella device fabrication process. We deposit a thick (3-4 μm) border of carbon around the sides of the sample before polishing it down to the desired thickness ($\sim 1 \mu\text{m}$). Then we add platinum to the corners to make electrical contacts. The thick carbon border electrically isolates the platinum electrodes from the sides of the Co₂MnGa crystal.

3 cont.) I understand this is a very tough experiment, do all of the six samples show exactly the same features?

The main result, a strong planar anisotropy with an unexpected 90° twist between the upper lower faces which is unaffected by external magnetic fields, invariably appears in each of the 6 samples we have studied (SM Fig. S2).

Furthermore, in each sample (see SM Table S4) the high-conductance-axis resistances match those expected for an *isotropic* sample of the same dimensions. As the thickness is increased (tested in samples up to 5- μm thick), these resistances sharply decrease, in agreement with the expectation for an isotropic metal slab. There is, however, some variation between samples in the magnitudes of the low-conductance-axis resistances. These range from ~ 0.7 to 4.4 Ω . However, in each sample these resistances are always much larger than the high-conductance-axis resistances. Furthermore, the anomalous resistances do not decrease with increasing sample thickness, in agreement with an anisotropy produced by surface states (see SM section D).

Reviewer #2 (Remarks to the Author):

The manuscript by N. Quirk et al. reports the observation of an unusual type of anisotropy in the electrical transport of a ferromagnetic Weyl semimetal, where the 90-degree rotation between the high-conductance axes of the top and bottom surfaces is documented, reproduced in multiple samples, and found to be consistent with C_{4v} symmetry. The weak magnetic field dependence of this anisotropy up to 9T, in contrast to the elimination of ferromagnetic domain walls at around 300e, suggests that this unusual behavior is independent of the domain wall conductance, but intrinsic to the electronic structure of the material. It is postulated to arise from surface states protected from strong hybridization with bulk states, and to support this hypothesis, an electrostatic finite element simulation has been used to attempt to reproduce the experimental results in a model consisting of a slab sandwiched between two ultrathin layers of anisotropic conductivity. To the best of my knowledge, no similar experimental results have been reported before, and it is a rather novel type of observation that serves a broad interest in the field of correlated electrons and topological materials. The experimental and analytical methods used are well established and explained. I would like to recommend this paper for publication in Nature Communication, but before doing so, I have some concerns that I hope the

authors will address.

Response to Reviewer #2:

Thank you very much for your insightful review of our paper. We have addressed each of your concerns to the best of our ability below.

1.

It is not clear to me how the x and y axes are defined and how this definition can be consistent across different samples and for comparison between electrical transport and MFM (Fig. 1d). The sample has cubic symmetry, while the x and y axes are along crystallographically equivalent directions (Table S4). In addition, according to the manuscript, the 90-degree rotation should be a spontaneous one reflecting the electronic instability (line 359). This suggests to me that it is equally possible for the top surface to have a high-conductance axis along x and along y , as well as for the bottom surface.

Your intuition is correct. Due to the cubic symmetry of the Co_2MnGa lattice, the $\langle 110 \rangle$ and $\langle -110 \rangle$ directions are equivalent and indistinguishable by X-ray scattering. In the revised versions of the manuscripts, we have removed all references to crystallographic directions.

The same 90°-twist anisotropy arises in thin lamellae cut from different source crystals and aligned to any of the crystal facets. In each sample we simply designated the x and y axes as the high-conductance directions on the upper and lower face, respectively. We have clarified this in the manuscript in lines 121-124 (marked version).

Our identification of the $\langle 110 \rangle$ and $\langle -110 \rangle$ axes in the MFM image of Fig. 1d was misleading. In the MFM study we arbitrarily labeled the horizontal direction as $\langle 110 \rangle$ and the vertical direction as $\langle -110 \rangle$. As mentioned in the text, the MFM lamellae do not have electrical contacts. We find that MFM imaging is not possible on samples with thick electrical contacts at the corners. In the new version of Fig. 1d we have removed the coordinate axes identifier because the x and y directions, which are defined by the anisotropy axes, have not been explicitly measured in these samples. Therefore, as we mention in lines 79-81 and the caption to Fig. 1, we do not know which axis aligns with the twisted Landau flux closure pattern, high- or low-conductance. This distinction is not needed to analyze the transport results. The 90°-twist anisotropy is unchanged in magnetic fields (measured up to 9 T) much stronger than those needed to completely erase the domain wall textures (< 1 T). We discuss this in the new version of the manuscript in lines 137-144 of the marked pdf. It would be interesting to conduct a future study focused solely on the twisted domain textures.

(1.1) However, in all samples measured, it is consistently reported that the top surface (with vertices 1,2,3,4) has high conductance along y , while the bottom surface (with vertices 1',2',3',4') has high conductance along x . Do the x and y axes have physical differences, or is it just a matter of practice to maintain consistent notation?

As mentioned, the x and y directions have been defined based on the observed high-conductance axes of the anisotropy in each sample. There are no obvious physical differences

between the two faces except that the lower face is in contact with the substrate. The thin lamella fabrication process is entirely symmetric in that the starting conditions have no physical differences between these two directions (different source crystals, arbitrary facets) and the sample is polished equally on the upper and lower faces.

(1.2) If it is a matter of notation, why do the R vs. T curves (Figure 1b and Figure S1) always show $R_{\{12,43\}}$ as monotonic while $R_{\{1'4',2'3'\}}$ is non-monotonic? This observation seems to suggest that surfaces with x - and y -directions as their high-conductance axes, or top and bottom (in contact with the SiO_2 substrate) ARE different, but why should they be?

In each sample the low-conductance axis on the lower face (y) has a non-monotonic RT with a large peak near 10 K and the low-conductance-axis on the upper face (x) has a quasi-logarithmic profile (steep downturn below 10 K). Indeed, this does indicate that the two low-conductance axes are not perfectly equivalent and that a weaker (sub-dominant) mechanism lifts the degeneracy between the poor-conducting axes on the upper and lower faces (it is weaker than the dominant one that drives the C_4 symmetry). We plan to investigate this weak symmetry-breaking effect in a later experiment.

The only obvious physical difference between these two configurations is that the lower face (non-monotonic RT) is resting on the substrate and the upper face is free, but it is not clear to us how this could change the temperature dependences. For example, contact with the substrate does not affect the lower-face high-conductance axis resistances, which have exactly the same temperature dependences of those of the upper face. Perhaps the mechanism that generates the low-conductance-axis resistances is more sensitive to the substrate, but we have no way of determining this. We have added a discussion of this higher-order symmetry breaking between the two low-conductance axes in the revised version of the main text (lines 208-212 of the marked pdf).

(1.3) For the MFM image shown in Figure 1d, if there is no in situ electrical transport measurement, how is it determined whether the Landau flux closure pattern is aligned with the high or low conductance axes? Alternatively, if there is always an EXPLICIT symmetry breaking between x - and y -directions or top and bottom surfaces, should any mechanisms be identified?

We have addressed this question in the first paragraph to this part (1), above. We do not know if the Landau flux-closure pattern is aligned with the high- or low-conductance axis, however, we emphasize that the domain textures have no impact on the transport anisotropy, which is unchanged in fields much higher than those required to erase the domain walls. We leave this to a future study. We have clarified this point in the revised fifth paragraph of the main text.

2.

The contact issue described in the Supplementary Materials, or whatever is causing the absence of low conductance in certain configurations, can be a major concern that potentially undermines the quality of the experiment. The absence of Ohm-scale resistance in some

configurations of sample N2 has been attributed to poor contact 3' (lines 77-85 in the Supplementary Texts). However, in Tables S2 and S3, configurations such as {1'4',2'3'}, {2'3',1'4'}, {33',22'}, {22',33'}, {11',2'3'} all involve contact 3' as either a current or voltage contact, and the greater-than-IsoEquiv resistance has all been identified. Why do the authors think that contact 3' specifically fails the measurements at {1'4', 33'}, {2'3',44'}, {33', 44'}, {33', 1'4'}, {44', 33'}, and {44', 2'3'}? Could there be other experimental problems? Will these factors affect the reported results in other configurations?

In the Supplementary Materials (SM), we have rewritten the section on Sample N2 (C) and added a new subsection (C.1) specifically discussing the issue with the magnitude of the anisotropy measured in the lower face. In this new section we explain how it appears that one of the lower-face contacts (3' or 4') does not properly sense the surface anisotropy. We actually do not know which of these two contacts is the culprit due to the C_{4I} symmetry (e.g., $R_{33',1'4'}$ and $R_{44',2'3'}$ are symmetric partners and both are too low to match the expected 90° twist anisotropy in the lower face). It appears that although this contact works (it never exhibits an open-circuit resistance), it does not sense the true anisotropy, as if the anisotropic area avoids it.

We suspect that, in this very thin sample, strain was induced at corner 3' or 4' while mounting onto the substrate. This corner is not capable of sensing the surface planar anisotropy; instead it only senses the bulk. Each lamella device takes upwards of 40 hours to fabricate so this small issue is quite unfortunate. However, we do not believe that it undermines the existence of a robust 90° twist in this sample. First, the upper face remains completely unaffected—it has a very clean planar anisotropy. Second, the high-conductance-axis resistance on the bottom face matches that of the top although is rotated by 90 degrees. It seems that the low-conductance-axis in the lower surface exist but is being ineffectively sensed by contact 3' or 4'. We conjecture that the reduced dimensionality of the anisotropic surface states may make them especially sensitive to disorder.

3. *In lines 185-189 it is mentioned that "although the raw transverse resistances, e.g. $R_{\{13,42\}}$ and $R_{\{4'2',3'1'\}}$, reflect the intrinsic 90° rotation anisotropy at zero H ", I understand that the authors are trying to attribute the raw resistance difference between $R_{\{13,42\}}$ and $R_{\{4'2',3'1'\}}$ to the 90° rotation, but this connection is not obvious. In fact, at zero field, if a rotation of $\pi/2$ is performed followed by an inversion with respect to the crystal center (C_{4I} as identified in the manuscript), the configuration $\{13,42\}$ is exactly mapped to $\{4'2',3'1'\}$. Do the authors mean that it is the fact that they are inequivalent that reflects the anisotropy? Then I don't follow the argument. Nevertheless, these planar transverse channels are not discussed at all in other parts of the manuscript, such as Figures 1, 2, 4, 5 and Tables S1, S2 and S3. They deserve a clearer explanation if the authors want to establish the trivial field-dependence of the anisotropy.*

Sorry for not explaining the Hall measurements. In the revised ms. we added a new paragraph on this topic (lines 240-275 in the marked pdf) and added a new panel (d) to Fig. 3. In all measurements of the raw transverse resistances, such as $R_{13,42}$, we observe the usual field-asymmetric term plus a much larger background term ΔR that is H independent and can be of

either sign. The raw curves are now displayed in the new Fig. 3d. As explained below, the large background shift ΔR (and its peculiar sign) is an unusual effect of the 90° twist anisotropy.

We have also changed the color scheme in the Hall plots (panels d-f) in order to clarify this connection. The following paragraph has been added to the main text.

The Hall curves reveal an unusual effect arising from the 90° -twist anisotropy. On each face we measure the field profiles of the 4 transverse resistances – $R_{13,42}$ and $R_{42,31}$ on the upper face and $R_{1'3',4'2'}$ and $R_{4'2',3'1'}$ on the lower. (In our sign convention $R_{ij,kl} > 0$ for holes in an isotropic square plate, implying $V_{kl} > 0$ with current source at i and drain at j). We find that, on the upper face, $R_{13,42}$ is comprised of a large H -independent (background) term ΔR that dwarfs its conventional H -antisymmetric profile. Surprisingly, its rotated partner $R_{42,31}$ shows the same weak H -antisymmetric term but its H -independent term is now negative (cyan curves in Fig. 3d for Sample O1 at 290 K). On the lower face, $R_{1'3',4'2'}$ and $R_{4'2',3'1'}$ exhibit H -antisymmetric terms closely similar to those in the upper face but they are now strongly shifted by large background terms $\Delta R'$ multiplied by an additional negative sign (grey curves). We may understand the origin of ΔR and $\Delta R'$ (including their signs) by examining the equipotential curves on the isotropic equivalents. On the upper face (left insert in Fig. 3d) the isotropic equivalent is lengthened along the less conducting axis \hat{x} . The sign of the potential drop $V_{42} > 0$ translates to a positive H -independent ΔR in $R_{13,42}$ (see equipotential curves). On the lower face (right insert) the sign of $V_{3'1'}$ (also positive) translates to a positive $\Delta R'$ in $R_{4'2',3'1'}$. For the rotated partners on either face, the signs invert. Hence we see that the large H -independent terms are inherent to the 90° twist. They cancel when we average the two curves on either face to yield the intrinsic Hall response R_{yx} .

4. *The simulation results presented in Figure 4 and Figure 5 do not seem to me to be dramatically different in their comparability to the experimental results, especially considering how the experimental results can vary in different samples (Table S4). Although I still think the COMSOL simulation is a valuable analysis, its support for the presence of surface states may not be as strong as one hopes. Even if the authors still want to advocate the surface state hypothesis, which I do not disagree with, I think alternative perspectives should also be offered.*

We have added a new section (G) to the SM titled “Alternate scenarios to the anisotropic surface-states picture” in which we describe 3 alternative models we considered that led us to converge to the thin-surface model. None of the previous models was able to come to close to recreating the experimental results. A key insight we made was that the condition for the high-conductance-axis resistances to match the expected values for the isotropic equivalents imposes a strict constraint: the majority of the sample volume has to be isotropically conducting with the conductivity of bulk Co_2MnGa . In each sample listed in Table S4, each $R_{14,23}$ agrees remarkably well to the isotropic-equivalent expectation. The high-conductance-axis resistance on the lower face, $R_{1'4',2'3'}$, does not always agree as well (e.g.,

Sample D1), but we expect this to be due to similar contact disorder effects in the lower as described for Sample N2. Additionally, the 1-Ohm magnitude of the z-axis resistances (e.g., $R_{11',22'}$) can only be achieved if the surfaces are quasi-separated from the bulk (large β) as mentioned, but also are extremely thin.

5. *There are a few minor problems with the manuscript.*

(5.1) *Figure 1b: It is not noted in which sample these T-dep curves are measured.*

We have identified these curves as being measured in Sample O1 in the new caption to Fig. 1.

(5.2) *Figure 2 has been referred to as Figure 3 throughout the manuscript and in the supplementary materials.*

We have corrected this.

(5.3) *Lines 253-254 give only the conductivity matrix for the top surface, not that of the bottom surface. I assume this is just a 90-degree rotation, but it should be noted.*

We have revised this Eq. 3 to include the conductivity for the lower surface. We have changed σ^s to σ^u and σ^l for the upper and lower surfaces, respectively. We have also changed the tensor definition in line 368 (marked pdf) to provide for both surfaces: $\sigma_{ij}^s \rightarrow \sigma_{ij}^{u(l)}$.

(5.4) *Figure 4d: Shouldn't there be a color bar indicating how the color scheme manifests the gradient of $V(r)$? I noticed that the caption says that the spacing between the contours is 0.01Ohm, but this does not help in the regions where the density of contours is high.*

Indeed. We have added a color bar to Figure 4d. In order to capture the few contours that extend all the way across the sample, we have limited the color range to ± 1 Ohm about the C_{4l} symmetric point in the simulation: the mid-point of the 33' edge. This is denoted by an arrow attaching the color bar to this midpoint and described in the new caption.

(5.5) *For the COMSOL simulation results shown in Figure 5, neither the caption nor the main text states what the optimal beta values are. This makes it difficult to compare with the results in Figure 4 and to evaluate whether the model used for the simulation remains reasonable when the delta increases dramatically.*

We have added the optimal β for the surface thickness dependence study to the caption of Figure 5. For $\delta = 10, 100, 300, 500$ nm the optimal β (to match $R_{12,43} = 1.9 \Omega$) were found to be 43100, 790, 110, and 30. The transition from anisotropic surface to bulk pictures can be seen in the drastic decline in the beta required to generate a 90° twist. (Increasing δ from 10 to 100 nm decreases β from 43100 to 790.) However, as mentioned in the main text (more explicitly in the new version), these anisotropies do not match the experimental results. Only as β is

dramatically increased ($\delta \leq 10$ nm) does the simulation approach experimental agreement. We see this as a cross-over to the described transfer matrix behavior.

(5.6) Figure S1: It is said that "the abrupt jump at $T = 120$ K in $R_{\{1'4',2'3'\}}$ was not seen in repeated cool-downs", then I suggest that a normal curve from one of the "repeated cool-downs" be shown instead of this one, which shows identified artifacts.

Unfortunately, although an RT without this jump was observed, additional full curves measured in Sample N2 from 300 K to 2 K could not be located. In the new manuscript, we point to the fact that Sample O1 (Fig. 1b) has no jump in $R_{1'4',2'3'}(T)$ to support our claim that this is not an intrinsic feature. We believe that this lower-face low-conductance-axis resistance jump could be caused by some settling while thermal cycling (it is a three-dimensional object rigidly fixed to a substrate). Regardless of its origin, the existence of this small jump does not significantly affect the transport properties of the device in any way. We have updated SM Section C and the caption to Fig. S1 to reflect this point of view.

(5.7) Table S4: Sample F1 also has an in-plane aspect ratio of 2:1, I am puzzled why the isotropic equivalent values for x and y directions do not differ, as they do in sample J1. Sample F1 is discussed in Supplementary Texts D2 for orientation dependence, where this aspect ratio difference is not mentioned.

There was a typo in Table S4 for the dimensions of Sample F1. It has a 1:1 aspect ratio, 10 x 10 x 1.2 μm . This has been corrected.

(5.8) Figure S4: In panel e, I suspect that the green curve should be $R_{\{12,1'2'\}}(H//x)$ instead of $R_{\{12,2'2'\}}(H//x)$ as labeled. I also wonder why its counterpart, i.e. $R_{\{12,1'2'\}}$ with $H//y$, is not shown in the same way as all the other three configurations.

We have changed it to $R_{12,1'2'}$. As mentioned in the caption, during the measurement of $R_{12,1'2'}$ with $\mathbf{H} \parallel \mathbf{y}$ in this sample, one of the contacts burned up due to an experimental mistake, preventing any additional measurements. We have also removed the non-antisymmetrized $R_{\{12,1'2'\}}$ with $H//y$ in panel c for this reason.

(5.9) Figure S5: 4'-3'-3'-2 should be 4'-3'-3-2.

Thanks, we have corrected this typo.

Reviewers' Comments:

Reviewer #1:

Remarks to the Author:

The manuscript was revised significantly according to the suggestion and some of my concerns were also clarified. Since the authors made these samples by FIB technique, to my knowledge, the FIB fabrication for small size samples always brought serious damages or pollution by Ga-ions injection or the formation of the insulating amorphous layer, especially on small sized samples. Based on my long-term experience by using FIB fabrication, transport properties on metallic quantum materials were always affected and shows very unexpected behavior mostly. I noticed that in the supporting material the authors made some descriptions on how to cut their samples, including using small beam current to polish the samples and covering the insulating carbon layer, I still doubt these protection can eliminate the influences of the Ga-injection. FIB cut has less effect on its magnetic properties, but usually dominates the transport properties. If all of the features observed here are intrinsic, it is absolutely very interesting and unusual, but the current data cannot strongly convince me to support their conclusion. Since the authors also made simulations, the results sounds to be consistent with the experimental data although it's not very solid.

I suggest the authors to make one more samples by using inert gas He or Ne FIB system, do the observed phenomena still survive? The inert gas FIB usually injects inert gas to samples but do not react with the samples.

Reviewer #2:

Remarks to the Author:

The revised manuscript has addressed all the concerns I raised in the review report. I have no more questions and would like to recommend its publication in Nature Communications.

*Response to Reviewers (Rev. 1 only)*Reviewer 1 comments:

The manuscript was revised significantly according to the suggestion and some of my concerns were also clarified. Since the authors made these samples by FIB technique, to my knowledge, the FIB fabrication for small size samples always brought serious damages or pollution by Ga-ions injection or the formation of the insulating amorphous layer, especially on small sized samples. Based on my long-term experience by using FIB fabrication, transport properties on metallic quantum materials were always affected and shows very unexpected behavior mostly. I noticed that in the supporting material the authors made some descriptions on how to cut their samples, including using small beam current to polish the samples and covering the insulating carbon layer, I still doubt these protection can eliminate the influences of the Ga-injection. FIB cut has less effect on its magnetic properties, but usually dominates the transport properties. If all of the features observed here are intrinsic, it is absolutely very interesting and unusual, but the current data cannot strongly convince me to support their conclusion. Since the authors also made simulations, the results sounds to be consistent with the experimental data although it's not very solid.

I suggest the authors to make one more samples by using inert gas He or Ne FIB system, do the observed phenomena still survive? The inert gas FIB usually injects inert gas to samples but do not react with the samples.

Rev. 1 response:

Thank you for reading the revised version of our manuscript and for your detailed comments. We agree that FIB polishing can leave a thin layer of disordered material on the crystal surfaces, regardless of how carefully the sample is prepared. However, our set up quite definitively excludes the possibility that the polishing process leads to the observed 90°-twist in the resistance anisotropy.

This is because the ion beam has a fixed direction relative to the sample throughout the polishing process. We have updated the Supplementary Information (section A.2, see changes marked in blue text) to make this clearer and more explicit. Let us call \hat{X} the direction of the ion beam relative to the sample (we reserve \hat{x} and \hat{y} for the observed electrical anisotropy). First, a thick slab (~10 x 10 x 10 μm) is cut from the source Co_2MnGa single crystal and lifted out. A thick carbon protection layer is added to each side face and then the slab is mounted on one side to a lift-out grid using in situ platinum deposition. The grid is put into a perpendicular holder, orienting the slab at an edge-on angle to the ion beam. The beam is then scanned line-by-line from the outside-in to polish the upper face by a set amount. Then the lower face is polished by the same amount, again from the outside-in. Crucially, both the upper and lower faces are polished with the beam in the same direction \hat{X} (the power, depth and time are also the same). If the milling process were to produce (say) a high-conductance direction, then *it must lie parallel to \hat{X} on both faces*. However, in all devices fabricated, our experiment uncovers anisotropy axes that are rotated by 90° between the two faces. We may safely exclude the possibility that the observed 90° rotation arises from the milling process.

Reviewers' Comments:

Reviewer #1:

Remarks to the Author:

The authors revised the manuscript and also made responses for my concerns. Actually, The authors did not directly response my concerns on how the transport data were affected by the Ga ions FIB fabrication for this small size devices, and also did not show the comparison on the samples fabricated using other ions beams, such as Xe or He. I already said the FIB may have little effect on the magnetic properties but have strong effect on its transport!

I understand the new experiments on other samples fabricated with Xe or He FIB system may take very long time and face difficulties, but it is really worth to do the comparison and to verify their conclusion in future.

Considering the authors have very rich experiences on the transport studies, I accept the authors explanations and recommend for acceptance for publication.

REVIEWERS' COMMENTS

Reviewer #1 (Remarks to the Author):

The authors revised the manuscript and also made responses for my concerns. Actually, The authors did not directly response my concerns on how the transport data were affected by the Ga ions FIB fabrication for this small size devices, and also did not show the comparison on the samples fabricated using other ions beams, such as Xe or He. I already said the FIB may have little effect on the magnetic properties but have strong effect on its transport! I understand the new experiments on other samples fabricated with Xe or He FIB system may take very long time and face difficulties, but it is really worth to do the comparison and to verify their conclusion in future.

Considering the authors have very rich experiences on the transport studies, I accept the authors explanations and recommend for acceptance for publication.

Our comments:

Thank you for your detailed comments and approval of our manuscript.

As we noted in our previous response, in all devices fabricated, our experiment uncovers anisotropy axes that are rotated by 90° between the two faces. Because the ion beam has a fixed direction relative to the sample throughout the entire polishing process, it cannot possibly generate the observed 90° rotation.